# Comparison of Experimental and Modeled EMI Shielding Properties of Periodic Porous xGNP/PLA Composites

**DOI:** 10.3390/polym11081233

**Published:** 2019-07-25

**Authors:** Avi Bregman, Eric Michielssen, Alan Taub

**Affiliations:** 1Department of Materials Science and Engineering, University of Michigan, 500 S State St., Ann Arbor, MI 48109, USA; 2Department of Electrical Engineering and Computer Science, University of Michigan, 500 S State St., Ann Arbor, MI 48109, USA

**Keywords:** graphene nanoplatelets, poly-lactic acid, compression molding, reflection loss, COMSOL, scattering parameters, EMI shielding, computational design

## Abstract

Microwave absorbing materials, particularly ones that can achieve high electromagnetic interference (EMI) absorption while minimizing weight and thickness are in high demand for many applications. Herein we present an approach that relies on the introduction of periodically placed air-filled pores into polymer composites in order to reduce material requirements and maximize microwave absorption. In this study, graphene nano platelet (xGNP)/poly-lactic acid (PLA) composites with different aspect ratio fillers were characterized and their complex electromagnetic properties were extracted. Using these materials, we fabricated non-perfect electrical conductor (PEC) backed, porous composites and explored the effect of filler aspect ratio and pore geometry on EMI shielding properties. Furthermore, we developed and experimentally verified a computational model that allows for rigorous, high-throughput optimization of absorbers with periodic porous geometries. Finally, we extend the modeling approach to explore the effect of pore addition on PEC-backed composites. Our composite structures demonstrated decreased fractions of reflected power and increased fractions of absorbed power over the majority of the X Band due to the addition of periodically arranged cylindrical pores. Furthermore, we showed that for xGNP/PLA composite material, reflection loss can be increased by as much as 13 dB through the addition of spherical pores. The ability to adjust shielding properties through the fabrication of polymer composites with periodically arranged pores opens new strategies for the modeling and development of new microwave absorption materials.

## 1. Introduction

With the advent of ubiquitous telecommunication networks and the expansive development of wireless electronics operating in the gigahertz (GHz) regime, “electromagnetic pollution” has grown to unprecedented levels [1]. Electromagnetic interference (EMI) arises when spurious or unintended electromagnetic (EM) signals radiated by electronic circuits interfere with the normal operation of surrounding equipment [2]. The standard approach to dealing with EMI is to prevent unintended radiation from reaching sensitive devices by encasing them in a metal cage or covering them with absorbing materials. Furthermore, since communication devices are increasingly multiband enabled, there has been a keen interest in broadband microwave absorbers with large and tunable absorption bandwidths [3,4,5].

Polymer composite foams represent an emerging approach to absorbance-dominated EMI shielding. When compared to non-porous polymer composites, foamed composites exhibit lower density, lower electrical percolation thresholds, and higher EMI shielding efficiency (*SE*) that is largely dominated by absorption [6,7,8]. The porous morphology decreases the impedance mismatch at the interface between air and the shielding material allowing a larger portion of the incident EM wave to penetrate the shield where it can be absorbed and dissipated [9]. Zhang et al. [10] reported a graphene-based foam with a reflection loss (RL) < −10 dB and an absorption bandwidth of approximately 60 GHz. In a similar vein, Kumar et al. [6] designed a magnetic nanoparticle decorated carbon polymer foam with an absorption bandwidth that covered entire X-Band (8–12 GHz). Finally, Li. et al. [11] designed phenolic foams with carbon nanotubes (CNTs) and iron oxide nanoparticles that demonstrated *SE* > 60 dB across the entire X-Band. While these foamed materials exhibit excellent shielding properties, their shielding characteristics depend highly on the foam morphology, which in turn is highly dependent on the foaming conditions [12,13,14]. For example, Gedler et al. [8] found that in a 0.5 wt % graphene nano-platelet (xGNP)/poly carbonate composite, there was a decrease in cell density accompanied by an increase in average cell size when changing the saturation temperature from 200 °C to 213 °C and the saturation pressure from 13.5 MPa to 16 MPa. This change in cell morphology in turn led to an increase in *SE* from 2 dB to 12 dB at 10 GHz. While foams make for good absorbers, the dependence of *SE* on their hard-to-control morphology makes development of new foam absorbers difficult.

Drawing inspiration from foams as well as metamaterials, “structural shields” have emerged as a new form of broadband microwave absorber. Luo et al. prepared honeycomb structures with an absorbing coating and were able to achieve an absorption bandwidth over the entire in the X-Band [15]. Similarly, Zhang et al. created a flexible reduced graphene oxide/polypropylene (RGO/PP) fabric that exhibited a minimum RL of −25 dB at 4 GHz [16]. By adding square pillars to the surface of the material, they were able to achieve an absorption bandwidth of 16 GHz as well as a RL of almost −50 dB at 9 GHz. Effects of pillar dimension were also investigated using computational tools. This hybrid computational/experimental approach allows for high throughput exploration of novel absorbing structures. While these pillared structures make for incredible absorbers, thicknesses in excess of 10 mm require significant amounts of material and prevent their use in devices with thickness requirements. 

In this study, we present a novel approach that relies on intrinsic electromagnetic properties to model the introduction of air-filled pores into conducting polymer composites to produce lightweight absorbers. In this study graphene nanoplatelet/poly-lactic acid (xGNP/PLA) composites with different aspect ratio fillers were characterized and their complex electromagnetic properties were extracted. Using these materials, we fabricated non-PEC backed, periodic porous composites and explored the effect of pore geometry and filler aspect ratio on EMI shielding properties. Furthermore, we developed and experimentally verified a COMSOL model that will allow for rigorous, high-throughput optimization of absorbers with periodic porous geometries. Finally, we extend the modeling approach to explore the effect of pore addition on PEC-backed composites. With a fully validated computational model, new materials can be quickly and thoroughly studied to produce fully optimized absorbers with periodic porous geometries.

## 2. Methods 

### 2.1. Composite Design Overview

Figure 1 shows the multistep design methodology for the design of optimized periodic porous EMI shields. In the first step (Figure 1A), polymer composite material is created through dispersion of xGNP into PLA and the complex electromagnetic parameters of the composite material are extracted using a Vector Network Analyzer (VNA). In the second step (Figure 1B), the complex electromagnetic parameters are used to simulate the scattering parameters or reflection loss of periodic-porous composites using COMSOL 5.3a^®^ [17]. In this step, geometric parameters such as the shape of a pore, position of a pore, thickness of the sample, thickness of the cell wall, and the number of vertically stacked pores, are modulated to identify optimal geometries that yield desirable EMI absorption. Following identification of optimized geometry, the final step (Figure 1C) consists of using compression molding and Computer Numeric Control (CNC) milling to fabricate composite samples with and without pores. The performance of fabricated samples is verified using a VNA. 

### 2.2. Materials and Procedures

Graphene nanoplatelets (xGNPs) with three different aspect ratios, M25 (D = 25 µm, t = 9 nm), M15 (D = 15 µm, t = 9 nm), and C750 (D =2 µm, t = 9 nm), were obtained from XG Sciences. Ingeo 4043D PLA pellets were purchased from 3DX. PLA was selected as the matrix polymer because future studies involving fused deposition modeling are planned. Methylene chloride (DCM) was supplied by Fisher Scientific. 

We used a conventional solution process to prepare composite material for step 1 (Figure 1A) [18]. 5 wt % loading. M25 Grade xGNPs (500 mg) were added to DCM and ultrasonicated for 1 h using a QSonica 700 probe (Qsonica, Newton, CT, USA) tip ultrasonicator to achieve dispersion. The same amount of energy is used for all composites, so they are assumed to have the same relative degree of dispersion. Then PLA pellets were added to the DCM dispersion and mixed overnight using a stir bar. Next the mixture was ultrasonicated for another 1 h. Finally, the mixture was cast into a large mold and left to dry overnight at ~110 °C. The resulting films were processed into smaller flakes using a grinder and compressed in a hot press at 230 °C and 20 MPa to obtain flat composite disks (Carver, Wabash, IN, USA). The aspect ratio of the xGNP was varied in the final composite.

The morphology of samples was observed using a JEOL 1500 scanning electron microscope (JEOL, Peabody, MA, USA). The samples were cryo-fractured in liquid nitrogen, and sputter coated with gold prior to visualization. The DC electrical conductivity of composites was determined using through thickness, two-point probe method on rectangular samples with a silver paint coating to ensure good electrical contact. A constant voltage of 5 V was applied to the samples and conductivity (*σ*) was calculated as
(1)$$\sigma =\frac{t}{\Omega wl}$$
where Ω is the electrical resistance, *l* is the length of the sample, *w* is the width of the sample, and *t* is the sample thickness. The EMI shielding effectiveness and complex electromagnetic parameters of xGNP/PLA composites were measured using an Anritsu MS4644B VNA (Anritsu, Atsugi, Japan). Prior to measurement, samples were cut and polished to achieve the desired cross-section for electromagnetic characterization using a standard X-band (8–12 GHz) waveguide setup (22.86 mm × 10.16 mm). A total of 1601 data points were taken in this frequency range for each sample. The permittivity and permeability of PLA composites were extracted from measured S-parameters using the Nicolson-Ross-Weir (NRW) method [19,20]. 

For step 2 (Figure 1B), Finite Element Modeling was performed in COMSOL. The modeled geometry consists of a 3D periodic Cartesian lattice with a unit cell that contains arbitrarily shaped air-filled pores inside of a material with pre-defined frequency dependent electromagnetic parameters. COMSOL’s Radio Frequency module is used to perform full field calculations of either a TE or TM incident wave that excites the structure from the top from a predefined angle of incidence (θ) (Port 1). Mesh refinement is used to maintain a minimum element quality of greater than 0.2 to ensure accurate results. The structure’s S11, S12, and reflection loss (RL) are calculated over the X-Band with a sampling interval of 0.1 GHz. A characteristic unit cell is shown in Figure 2A. The unit cell repeats periodically in the x-y plane and has finite thickness in the z-direction. The top of the cell is terminated by a perfectly matched layer (PML) to prevent secondary reflections, and the bottom of the cell is terminated either by a perfect electrically conducting (PEC) or a second port for study of non-PEC backed composites.

To prepare samples for step 3, xGNP/PLA composites with a periodic array of cylindrical pores (Figure 2B) are fabricated using a Tormach PCNC440 milling machine (Tormach, Waunakee, WI, USA) and their EMI shielding properties were characterized using a similar setup in step 2. Measured EMI shielding properties were compared to simulated values from step 2 for model validation.

## 3. Results and Discussion

### 3.1. Microstructure and Conductivity of xGNP/PLA Composites

Figure 3 shows representative SEM images of the cross section of xGNP/PLA composites with different aspect ratio fillers. We have observed a degree of agglomeration which is attributed to long solvent evaporation times giving fillers time to coalesce. There is also a degree of filler alignment that can be seen, particularly in Figure 3C. This can be ascribed to the compression molding that is used to form composite plaques. From the SEM image of C750/PLA (Figure 3B) we can clearly see that there is a large separation between filler material as compared to the other composites. For this specimen we expect a decrease in the inter-nanostructure connections that will ultimately play an important role in the EMI shielding characteristics of the composite material.

Table 1 shows the DC electrical conductivity of 5% xGNP/PLA composites with different aspect ratio xGNPs at room temperature. As the aspect ratio of the filler material increases, there is an increase in conductivity from 6 × 10^−10^ for C750 filler material to 1.5 × 10^−1^ for M25 filler material. It is well known that increases in aspect ratio decrease the percolation threshold, thus making it easier to form conductive networks [21]. In addition, the increase in maximum electrical conductivity also increases with higher aspect ratio [22].

### 3.2. Complex Electromagnetic Properties vs. xGNP Loadings and Frequency

The permeability of PLA composites is not measurably affected by the addition of xGNP since there is no magnetic component in the filler material and any effects are likely due to measurement error. On the other hand, there is a measurable effect in the complex permittivity due to xGNP loading. The permittivity of neat polymers is usually low but can be enhanced through the addition of conductive filler material; the polarization of the matrix, polarization at matrix/filler interfaces, and the polarization of conductive filler material all make contributions to the final composite permittivity [23]. 

Figure 4 shows the experimentally derived real ($\varepsilon^{\prime }$) and imaginary ($\varepsilon^{\prime \prime }$ portions of the complex relative permittivity as a function of frequency for the xGNP/PLA composite plaques. The real part is related to the storage capacity of electric energy mainly due to the degree of polarization within the material [24]. The imaginary permittivity is the associated degree of energy dissipation from Joule heat generated during conductance and thermal energy due to relaxation loss mechanisms [25]. The real permittivity is sensitive to the type of xGNP filler. The real permittivity of unmodified PLA is ~2.8 across the entire frequency range and increases to ~17.6 for 5% M25 loading. Previous studies have shown that if the filler concentration is low, there is not a measurable effect on ε’ with respect to frequency. In the case of our xGNP/PLA composites there is minimal frequency dependence in the X-Band. It is known that for ellipsoidal particles, the percolation threshold decreases with increasing aspect ratio [26]. Therefore, at a fixed loading, higher aspect ratio xGNP fillers will have more conducting pathways and display higher values for real permittivity due to increased capacitor like interactions between filler particles [27,28].

With increasing aspect ratio of xGNP filler, there is also an increase in the imaginary part of the permittivity. This can be attributed to enhanced polarization loss and higher hopper conductivity due to enhanced filler interactions [29,30]. The 5% M25/PLA and 5% M15/PLA samples exhibit slightly decreasing ε’’ with increasing frequency, but the 5% C750/PLA displays a distinct relaxation peak [31].

### 3.3. EMI Shielding Effectiveness of Non-PEC Backed xGNP/PLA Composite Plaques in a Waveguide

The effect of different grades of graphene in xGNP/PLA on electromagnetic interference shielding effectiveness (*SE*) was investigated. EMI shielding refers to reflection and absorption of incident electromagnetic radiation by a material. The efficiency of a shielding material is dependent on the frequency of incident EM radiation, shield thickness, as well as the intrinsic electromagnetic properties of the material [32]. EM wave propagation in any medium is dependent on the impedance of the medium (Z = √(μ/ε)) as well as the EM wave’s velocity inside of that medium (υ = 1/√με) [33]. When a traveling wave encounters an interface with different impedances on either side, there will be a change in wave velocity as well as partial reflection with the magnitude of this interaction determined by the difference in impedance between the two mediums [34]. Any changes in permittivity or permeability will impact a materials response to electromagnetic radiation, particularly if that material is a nonconductive polymer matrix. 

The total EMI shielding effectiveness (*SE_T_*) is defined as.
(2)$$SE_{T}\;\left(dB\right)=-10\log_{10}\left({\left|S_{12}\right|}^{2}\right)$$

The mechanisms of EMI shielding can be defined by the fraction of incident power that is reflected (R = |*S*_11_|^2^ = |*S*_22_|^2^), transmitted, (T = |*S*_12_|^2^ = |*S*_21_|^2^), and absorbed (A = 1 – T − R):

Figure 5 shows the *SE_T_* and power balance for 3 mm thick PLA composites with different aspect ratio xGNPs measured in a waveguide over the X-Band. As illustrated in Figure 5A, *SE_T_* increases with increasing xGNP aspect ratio over the entire frequency range with a minimum of 12 dB at 8 GHz for the M25/PLA composite. The observed trends and measured *SE_T_* for xGNP/PLA composites resemble those of xGNP/PBAT composites with similar complex electromagnetic parameters [35]. For M15 and M25 composites, *SE_T_* decreases with increasing frequency, which is a phenomenon that has been observed in other conductive polymer composites [36,37]. C750 and PLA exhibited minimal frequency dependence for *SE_T_* in this frequency range. Yang et al. observed the *SE* of carbon-non fiber/polystyrene foams and found that at loadings of 10% and higher there was a dependence on frequency but did not see the same effect with lower loadings of carbon nanofiber [37].

The power balances for xGNP/PLA composites with different aspect ratio fillers are plotted in Figure 5B. For neat PLA composites, most of the incident EM radiation transmits through the material; there is a portion of the power that is reflected due to slight impedance difference with air, and almost no incident radiation is absorbed. For all grades of xGNP, the transmission coefficient is lower than that of neat PLA. Over the entire investigated range, higher aspect ratio xGNP fillers display low levels of transmission and high levels of reflection; as permittivity of nanocomposites increases without a change in permeability, the higher impedance mismatch at the interface leads to increased reflection. It can be concluded from these data that reflection is the primary shielding mechanism for xGNP/PLA composites in the X-band. This agrees with similar conductive polymer composites systems [38,39,40].

### 3.4. EMI Shielding Effectiveness of Non-PEC Backed PLA/xGNP Composites with Cylindrical Pores

For non-PEC backed shields, the pursuit of higher conductivity and higher impedance match is an ideal strategy for achieving high-performing, reflection-dominated EMI shields. However, high performing EMI absorbers rely on complementary effects between dielectric loss, magnetic loss, conductivity, and impedance matching [3,15,41,42]. Controlling impedance mismatch while simultaneously ensuring there are enough pathways for EM dissipation can be a difficult task to achieve. As previously discussed, one method for achieving this balance is using foamed composites. By introducing air in the form of a highly porous morphology, the *SE_T_* of foamed composites can be greatly improved as compared to their non foamed counterparts due to decreased impedance mismatch at the surface as well as increased interactions at pore-matrix interfaces inside of the foam [23,43]. However, the inability to model the shielding efficiency of foamed materials a priori limits their development to trial and error experiments that are costly in terms of time and resources. 

In order to control impedance matching we propose using an arrangement of periodic pores that can be explicitly determined beforehand. Because the pores are defined independently of materials processing, there is added flexibility that can be given in terms of the size, shape, and position that pores can have. To demonstrate this point, the effect of adding cylindrical pores with a constant radius and varying height in a periodic layout was explored and the corresponding geometries were modeled in COMSOL to demonstrate the ability to successfully model the structures.

Figure 6 shows the measured scattering parameters and power balance as measured in a waveguide for 4.6 mm thick 5 wt % M25/PLA composites with and without cylindrical pores of varying height. The trend for the modeled total shielding efficiency follows that of the experimental data illustrating the model’s utility in guiding the selection of pore geometry. The non-porous composite exhibits a maximum *SE_T_* of 10.9 dB at 11.6 GHz. With the addition of a single layer of periodic cylindrical pores, there is a monotonic decrease in *SE_T_* as well as a gradual shift of minimum *SE_T_* to higher frequencies with increasing pore height. The addition of pores also has a measurable effect on the power balance as shown in Figure 6B. As pore height increases there is a noticeable decrease in the reflection coefficient at lower frequencies as compared to the sample with no pores. This can be attributed to the increased impedance matching due to the addition of air to the sample. Increased interaction with conductive pathways as well as increased internal reflections at intra-composite interfaces leads to an increase in absorption coefficient over the entire frequency range for porous samples. However, 5 wt % M25/PLA has not been optimized for high shielding efficiency. There is a noticeable increase in the transmitted power fraction, with as much as 30% of incoming radiation transmitting for the sample with 4 mm pores. For this material, the addition of pores leads to higher absorbed power fractions, but lower *SE_T_* as compared to non-porous samples.

As mentioned above, quick development of foamed composites is limited due to lack of computational tools for modeling morphology as well as the *SE* due to unique foamed morphologies. To demonstrate the validity of a model guided approach to periodic porous geometries, we also simulated these periodic structures in the COMSOL environment using the measured electromagnetic properties for 5% M25/PLA. Overlaid against the measured results in Figure 6, there is general agreement in the trends between the simulated and measured scattering parameters and power balance for cylindrical pore examples. The lack of absolute correlation can be attributed to differences in the actual sample such as pore mismatch when aligning the two separate halves or precision errors that can occur during machining. 

This process was repeated for 5% M15/PLA composites and the resulting comparison between experimental and simulated scattering parameters and power balance is shown in Figure 7. It can again be seen that there is good agreement between the trends for experimental and simulated data. There is also a non-monotonic increase in *SE_T_* with an increase in pore height for 5% M15/PLA samples that is worth noting because it shows that a modeling approach is an effective method for identifying non-linear trends.

### 3.5. Reflection Loss of PEC-Backed xGNP/PLA Composites

In many applications involving EMI shields, the absorbing material is backed by highly reflective metallic casings. For this purpose, it is also informative to explore materials backed by a PEC instead of air. We showed in previous simulations that for a composite made of 30 wt % graphene/Fe3O4 paraffin wax composite, the addition of a single spherical pore with a radius of 1.25 mm could cause as much as a 13 dB increase in RL [44]. Figure 8 shows the results of COMSOL simulations demonstrating the effect of adding either a spherical pore (A) or a cylindrical pore (B) to a 4 mm thick sample with the complex electromagnetic properties of 5% M25/PLA. The addition of a single pore, when optimized, can have a pronounced effect on the RL; a single spherical pore with a radius of 1.2 mm can lower RL from −32 dB to −45 dB at a frequency of 9.2 GHz. The addition of cylindrical pores can also have a measured effect on RL, but there are subtle differences in the position of peaks as compared to the spherical pore. Particularly for a radius of 1.6 mm, the minimum peak position varies by almost one GHz. The improvements in RL as well as the different effects of different geometries reinforces the need for model guided experimentation for EMI shielding materials.

## 4. Conclusions

The connection between experimental and simulated data for xGNP/PLA composites illustrates the promise of a modeling guided approach to pore addition starting from known electromagnetic parameters. The morphology, conductivity, and electromagnetic properties of xGNP/PLA composites with different aspect ratio xGNP fillers were explored. It was found that SE_T_ in the X-band improves with higher aspect ratio fillers at a fixed weight percentage. By analyzing the shielding coefficients, it was shown that reflectance is the dominant shielding mechanism due to large impedance mismatch at the front face. In addition, the effect of adding periodically arranged cylindrical pores on the EMI shielding properties was explored. The addition of pores had the desired effect of increasing absorption while decreasing reflection but increased the transmission as well. The porous geometries were simulated in COMSOL and showed excellent agreement with experimental data. The ability to adjust shielding properties through the fabrication of conductive polymer composites with periodically arranged pores opens new strategies for the modeling and development of new microwave absorption materials.

## Figures and Tables

**Figure 1 polymers-11-01233-f001:**
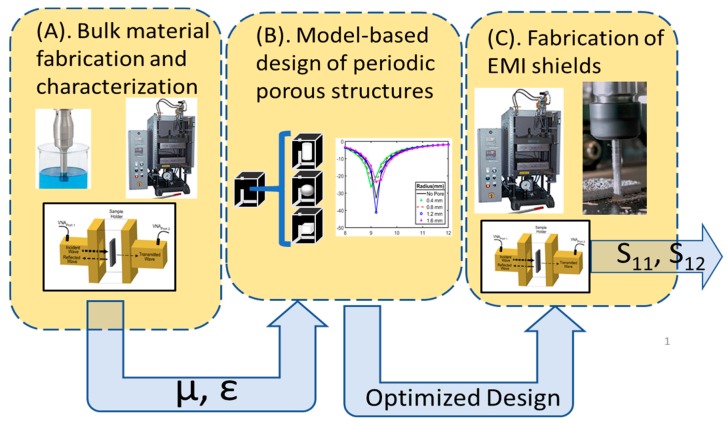
Three step design methodology for new EMI shields. (**A**) Composite materials are fabricated through a combination of ultrasonication and compression molding. Complex electromagnetic properties are extracted using a VNA. (**B**) Measured complex electromagnetic parameters are used in the COMSOL environment to optimize periodic porous geometry. (**C**) Optimized geometries are fabricated using a combination of compression molding and CNC milling. Final EMI shielding properties are measured and tested once more in a VNA.

**Figure 2 polymers-11-01233-f002:**
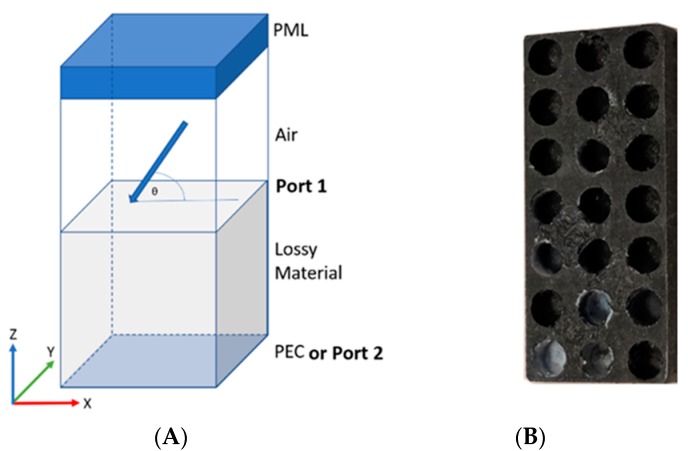
(**A**) Schematic of unit cell used in COMSOL simulations. The top of the cell is truncated with a perfectly matched layer (PML) to prevent reflections at the top boundary. EM waves with incident angle θ are generated from Port 1 and are incident upon a lossy material with predefined permittivity and permeability. The bottom of the cell is either backed by a perfect electrical conductor or a second port. The unit cell is periodic in the x/y direction and finite in the z direction. (**B**) A representative xGNP/PLA sample after being machined. The periodic porous array is milled on two separate halves which are then bonded together to form completely encapsulated pores.

**Figure 3 polymers-11-01233-f003:**
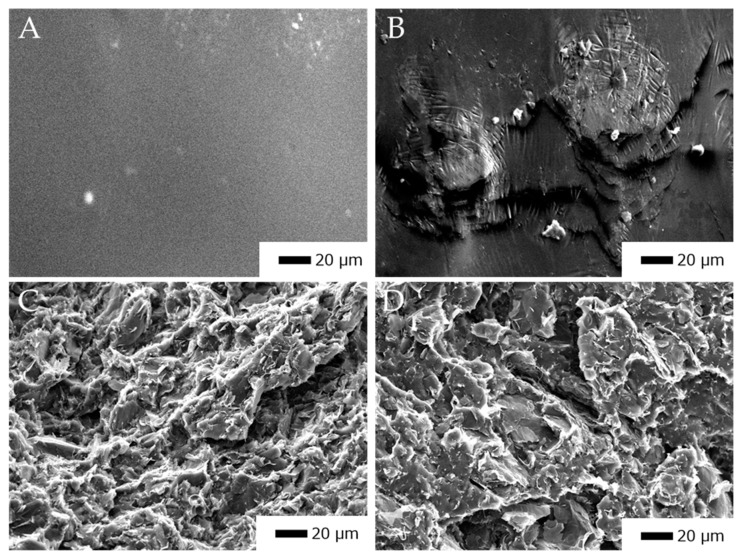
Representative SEM images of (**A**) PLA (**B**) 5% C750/PLA (α = 222), (**C**) 5% M15/PLA (α = 1667), and (**D**) 5% M25/PLA (α = 2778) composites.

**Figure 4 polymers-11-01233-f004:**
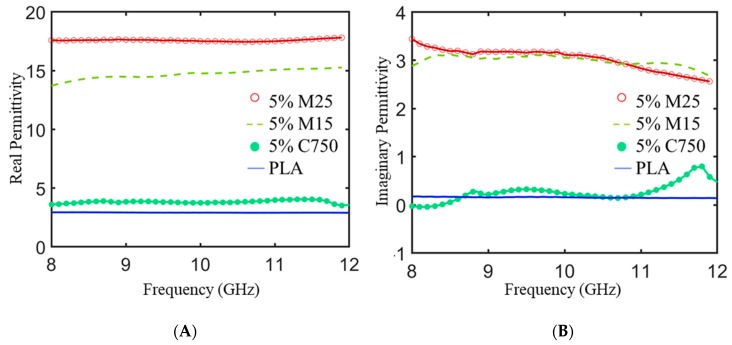
The real part (**A**) and imaginary part (**B**) of the complex permittivity for 5 wt % xGNP/PLA composites fabricated with different aspect ratio filler material. The properties of neat PLA are also displayed as reference.

**Figure 5 polymers-11-01233-f005:**
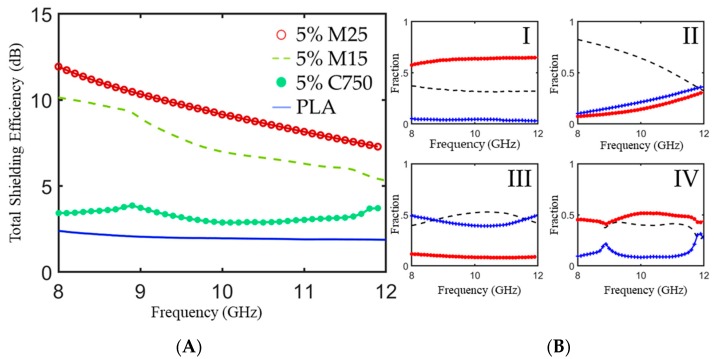
(**A**) Total shielding efficiency as measured in an X-band waveguide for xGNP/PLA composite plaques fabricated with different aspect ratio starting material. (**B**) Power balance for 5 wt % xGNP/PLA composite plaques loaded with (I) neat PLA, (II) M25, (III) M15, and (IV) C750. All plaques have a thickness of 3 mm.

**Figure 6 polymers-11-01233-f006:**
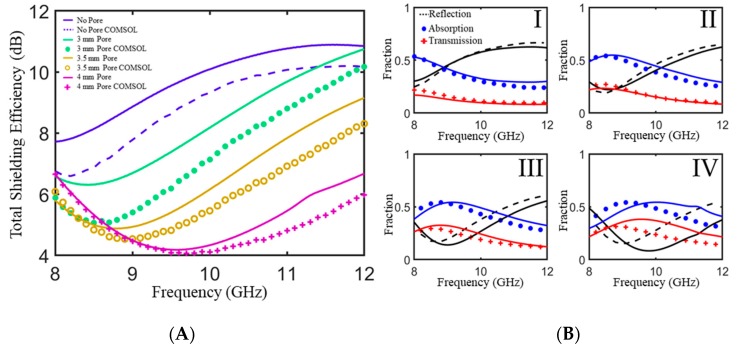
(**A**) Measured and simulated total shielding efficiency in an X-band waveguide for a 5% M25/PLA composite plaque with and without pores of varying height. (**B**) Measured and simulated power balance for a 5% M25/PLA composite plaque (I) without pores, (II) a cylindrical pore with a height of 3 mm, (III) a cylindrical pore with a height of 3.5 mm, and (IV) a cylindrical pore with a height of 4 mm. All plaques have a total thickness of 4.6 mm. All cylindrical pores have a diameter of 2.54 mm. Simulated data is depicted with symbols and experimental data is represented by solid lines of the same color.

**Figure 7 polymers-11-01233-f007:**
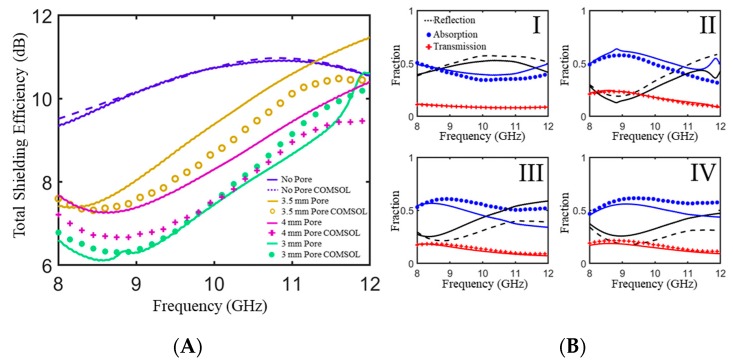
(**A**) Measured and simulated total shielding efficiency in an X-band waveguide for a 5% M21/PLA composite plaque with and without pores of varying height. (**B**) Measured and simulated power balance for a 5% M15/PLA composite plaque (I) without pores, (II) a cylindrical pore with a height of 3 mm, (III) a cylindrical pore with a height of 3.5 mm, and (IV) a cylindrical pore with a height of 4 mm. All plaques have a total thickness of 4.6 mm. All cylindrical pores have a diameter of 2.54 mm. Simulated data is depicted with symbols and experimental data is represented by solid lines of the same color.

**Figure 8 polymers-11-01233-f008:**
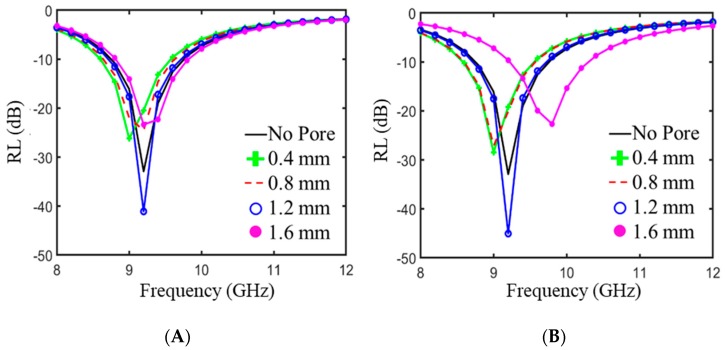
Modeled reflection loss for 5%M25/PLA with thickness of 4 mm and modified with either a single spherical pore (**A**) or a cylindrical pore (**B**). Both plots show the effects of a changing radius. In the case of the cylindrical pore, the height is fixed at 1.8 mm.

**Table 1 polymers-11-01233-t001:** Measured conductivity of xGNP/PLA composites and the reported aspect ratio of xGNP filler material.

Material	Conductivity (S/m)	Aspect Ratio
PLA	10^−13^	N/A
5% C750/PLA	6 × 10^−10^	222
5% M15/PLA	3 × 10^−4^	1667
5% M25/PLA	1.5 × 10^−1^	2778

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
