# Peer review of "Comparison of Experimental and Modeled EMI Shielding Properties of Periodic Porous xGNP/PLA Composites"

_polymers, 2019, doi:10.3390/polym11081233_

Round 1

Reviewer 1 Report

I my opinion the paper Comparison of Experimental and Modeled EMI 3 Shielding Properties of Periodic Porous xGNP/PLA 4 Composites is well written and could be published in Polymers after small correction.

1/ Authors shouldn't use  "The real part (a) and imaginary part (b)" should be experimental and modeled. 

2/ SEM pictures of different materials should be in the same scale because in current version it is difficlt to compare.

3/ Some misstyping should be improved e.g. doubled spaces or lack of them. 

Reviewer 2 Report

1.      The full name of PLA (Line 16: polyactic acid ) and PEC ( Line 18: perfect electrically conducting ) needed to be demonstrated in the abstract.

2.      The authors mentioned conductive polymer composite in line 43, however, the name of this material is not seen.  Is it polyaniline? And how conducting polymer is fabricated into the composite?

3.      Two-point probe ( line 121 ) is not as accurate as four-point probe in measuring conductivity?

4.      Why choose polylactic acid ( PLA ) as the binding material for xGNP as EM wave absorber? Why not mix polyaniline with xGNPs instead of PLA?

5.      There is no discussion about the foamed conducting polymer composite with different shapes of pores inside. Only xGNP/PLA with different aspect ratios are prepared and discussed.

6.       The caption of Figure 3 is missing in page 5 of 13.

7.      Line 171. The ‘read’ should be ‘red’.

8.      In Figure 5(a), why COMSOL always gives positive deviation from the experimental data?

9.      Why M25/PLA with no pore owns the best shielding effectiveness found in Figure 5(a)?

10.  The data of reflection loss can be obtained directly from instruments and compared with the modeled results in Figure 8.

Reviewer 3 Report

The manuscript submitted by Bregman et al. investigated the filler effect of aspect ratio and pore geometry to EMI shielding properties of graphene nano platelet (xGNP)/PLA composites by experiment and modeling. However, the authors need to revise some of the experimental sections. From the SEM, it’s difficult to distinguish the filler, which is due to low resolution images of graphene nanoplatelet and PLA matrix. The conductivity of the composites was discussed from aspect ratio in Table 1. However, the dispersion of fillers in PLA will also affect the conductivity. In addition, the neat PLA polymer should be characterized in SEM and conductivity as a control.

Round 2

Reviewer 2 Report

Is polyaniline  the conductive polymer used in the studies?   How conductive polymer is fabricated into the composite?

Author Response

We do not use polyaniline, or any other conductive polymer. We use PLA as our polymer matrix, and electrical conductivity comes from the addition of graphene nanoplatelets. To avoid confusion, we have modified the first paragraph on page 2 to say

“Polymer composite foams represent an emerging approach to absorbance-dominated EMI shielding. When compared to non-porous polymer composites, foamed composites exhibit lower density, lower electrical percolation thresholds, and higher EMI shielding efficiency (SE) that is largely dominated by absorption [6–8].”

The fabrication of our composites is discussed in detail in the materials and methods starting on line 112.

Reviewer 3 Report

Accept in Polymers

Author Response

Thank you